# Comparative Investigation on the Emission Properties of $RAl_3(BO_3)_4$ (R = Pr, Eu, Tb, Dy, Tm, Yb) Crystals with the Huntite Structure

**Enrico Cavalli** [1],*  **and Nikolay I. Leonyuk** [2]

[1] Department of Chemical Sciences, Life and Environmental Sustainability, Parma University, 43124 Parma, Italy

[2] Department of Crystallography and Crystal Chemistry, Lomosonov Moscow State University, Moscow 119991, Russia; leon@geol.msu.ru

* Correspondence: enrico.cavalli@unipr.it; Tel.: +39-0521-905436

**Abstract:** The luminescence properties of $RAl_3(BO_3)_4$ (RAB, with R = Pr, Eu, Tb, Dy, Tm, Yb) huntite crystals grown from $K_2Mo_3O_{10}$ flux were systematically characterized in order to investigate their excitation dynamics, with particular reference to the concentration quenching that in these systems is incomplete. To this purpose, selected excitation, emission, and decay profile measurements on diluted R:YAB crystals were carried out and compared with those of the concentrated compounds. The effects of the energy transfer processes and of the lattice defects, as well as the ion-lattice interactions, have been taken into consideration in order to account for the experimental results.

**Keywords:** borate crystals; luminescence; rare earth spectroscopy

## 1. Introduction

Borate crystals of the huntite family with the general formula $RX_3(BO_3)_4$ (R = lanthanide ions, X = Al, Ga, Fe, Cr) have been the subject of both fundamental and technologically oriented studies [1,2] for more than fifty years. The ability to grow good optical quality single crystals at relatively low temperatures [3], the insensitivity of the huntite structure to the cation replacement, and the presence of a single lanthanide site with defined local symmetry make these materials very suitable for investigations of the structure of the energy levels and the de-excitation mechanisms of the active ions [4,5]. Furthermore, the combination of favorable chemical and physical characteristics like stability, hardness, high UV transparency, and nonlinear optical properties make them attractive for application in several fields: lasers [6,7], scintillators [8], phosphors [9,10] and so on. In general, these crystals require active media, usually constituted by a transparent host matrix, like $YAl_3(BO_3)_4$ or $YGa_3(BO_3)_4$, containing luminescent ions in low amounts. This not only for cost reduction purposes, but also because high concentrations of active ions can result in emission quenching effects, a consequence of energy transfer processes whose efficiency depends on the ion nature and on the characteristics of the host. For these reasons, research activity has mainly focused on doped materials ($YAl_3(BO_3)_4$:$R^{3+}$, hereafter R:YAB, is the most popular) and less attention has been dedicated to the study of the spectroscopic properties of concentrated compounds. Nevertheless, the limited number of investigations carried out on these materials has provided interesting information concerning the effectiveness and the mechanisms of the excitation transfer [11,12], the effect of the rare earth substitution on the structural properties [13], and the development of microchip lasers [14,15]. Consequently, we felt it would be interesting to revisit part of the existing literature and to extend it to unexplored members of the $RAl_3(BO_3)_4$ (RAB) family, in order to provide a general picture of their emission properties and of the effects of concentration and structure on their luminescence performances.

## 2. Materials and Methods

### 2.1. Crystal Growth and Structural Properties

The RAB and R:YAB crystals (R = Pr, Eu, Tb, Dy, Tm, Yb) were grown from $K_2Mo_3O_{10}$–based flux melts in the 1150–900 °C temperature range. The details of the growth procedure are well described in several papers [3,5,13]. The flux growth technique, adopted because the RAB compounds melt incongruently, entails the unavoidable contamination of the crystals by flux components. Investigations in this area have demonstrated that only Mo ions in the tri-, penta- or hexavalent oxidation state are present at a level of some relevance (0.1–0.5%) [1]. The crystals used for the spectroscopic experiments are in the form of rods up to $2 \times 1 \times 1$ mm$^3$ in size and free from inclusions. Based on X-ray powder diffraction studies [16], the first twelve $RM_3(BO_3)_4$ borate crystals synthesized by Ballman in 1962 [17] were classified structurally as part of the huntite family, $CaMg_3(CO_3)_4$, $R32$ space group (Table 1).

**Table 1.** Crystallographic characteristics of $RAl_3(BO_3)_4$ (space group $R32$) [1].

| Borate | a (Å) | c (Å) |
|---|---|---|
| $YAl_3(BO_3)_4$ | 9.288(3) | 7.226(2) |
| $PrAl_3(BO_3)_4$ | 9.357(3) | 7.312(3) |
| $EuAl_3(BO_3)_4$ | 9.319(3) | 7.273(3) |
| $TbAl_3(BO_3)_4$ | 9.297(3) | 7.254(3) |
| $DyAl_3(BO_3)_4$ | 9.300(3) | 7.249(3) |
| $TmAl_3(BO_3)_4$ | 9.282(3) | 7.218(3) |
| $YbAl_3(BO_3)_4$ | 9.278(3) | 7.213(3) |

Unlike the carbonate mineral structure, which has only trigonal modification (space group $R32$), the rare-earth-aluminum borates containing large $R$-cations are represented by three polytypic modifications with space groups $R32$, $C2/c$ and $C2$ (Table 2).

**Table 2.** Unit cell parameters of monoclinic $RAl_3(BO_3)_4$ modifications.

| Borate | a (Å) | b (Å) | c (Å) | β, degree | Sp. Gr. | Ref. |
|---|---|---|---|---|---|---|
| $PrAl_3(BO_3)_4$ | 7.272(2) | 9.362(2) | 11.145(3) | 103.49 | C2/c | [18] |
| $EuAl_3(BO_3)_4$ | 7.270(4) | 9.328(6) | 11.074(4) | 103.17 | C2/c | [18] |
| $EuAl_3(BO_3)_4$ | 7.230(2) | 9.322(4) | 16.211(4) | 90.72(2) | C2 | [19] |
| $TbAl_3(BO_3)_4$ | 7.220(3) | 9.312(4) | 11.072(4) | 103.20(3) | C2/c | [19] |

In previous studies [19,20], these polytypes are described in terms of the OD-theory. In this case, monoclinic polytypes are represented in accordance with the transformation of rhombohedral $R$-cell to monoclinic $C$-cell with $β = 113°$ for both $C2/c$ and $C2$ space groups. The crystals investigated in this work belong to the R32 polytype. In the hexagonal huntite lattice, the $R^{3+}$ sites have six-fold oxygen coordination and trigonal prismatic geometry with $D_3$ point symmetry. The $Al^{3+}$ ions occupy octahedral sites and the $B^{3+}$ ions are surrounded by three oxygen atoms with triangular geometry (Figure 1a).

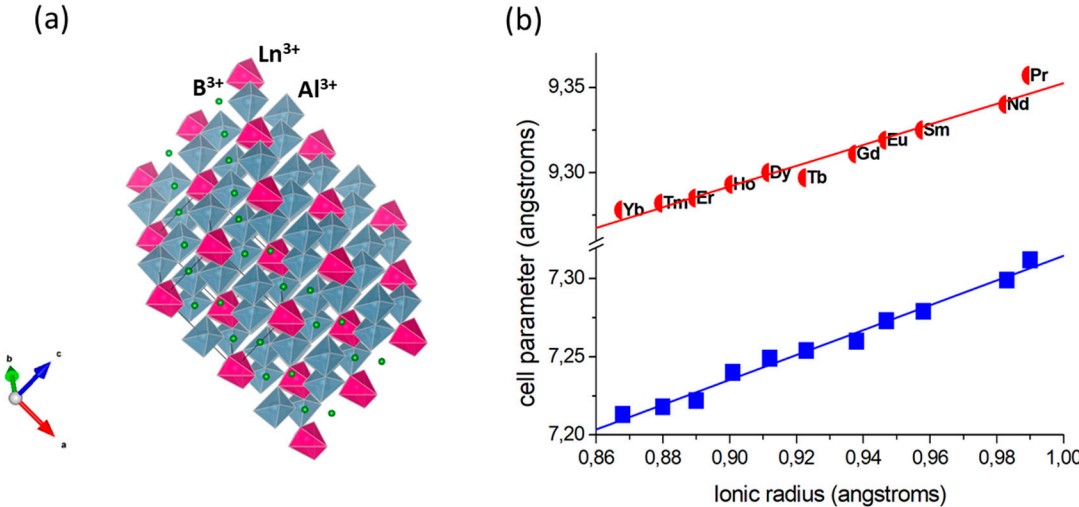

**Figure 1.** (**a**) LnAB crystal structure (elaborated using the VESTA software [21]); (**b**) variation of the cell parameters (taken from Reference [1]) as a function of the $Ln^{3+}$ ionic radii (from Reference [22]).

The $Ln^{3+}$ sites are well separated from one another, the $R^{3+}$–$R^{3+}$ minimum distance being of the order of 5.8–5.9 Å. This limits energy transfer and concentration quenching processes. It is interesting to note that the cell parameters increase linearly with the ionic radius of the rare earth ions. This dependence can be formalized by the following equations:

$$a\left(\mathring{A}\right) = 8.75 + 0.61 \cdot r\left(\mathring{A}\right) \tag{1}$$

$$c\left(\mathring{A}\right) = 6.52 + 0.79 \cdot r\left(\mathring{A}\right) \tag{2}$$

It would be interesting to verify if this model could be extended to other members of the huntite family. Despite their phenomenological nature, they can be useful in different circumstances, like in predicting lattice parameters of unknown compositions, estimating thermodynamic properties, testing data, etc. [23].

### 2.2. Spectroscopic Measurements

The emission spectra and the decay profiles were measured at room temperature using an Edinburgh FLS1000 (Edinburgh Instruments, Livingston, UK) or a Jobin-Yvon FluoroMax 3 spectrofluorimeter (Horiba, Kyoto, Japan).

## 3. Results

### 3.1. PrAB

The emission properties of PrAB were investigated by Koporulina [24] and Malyukin [12]. They observed two band systems centered at 610 and 650 nm, ascribed to the $^1D_2 \rightarrow {}^3H_4$ and $^3P_0 \rightarrow {}^3H_6$ transitions, respectively. An accurate inspection of the spectra shown in Figure 2a allows the assignment of some additional transitions.

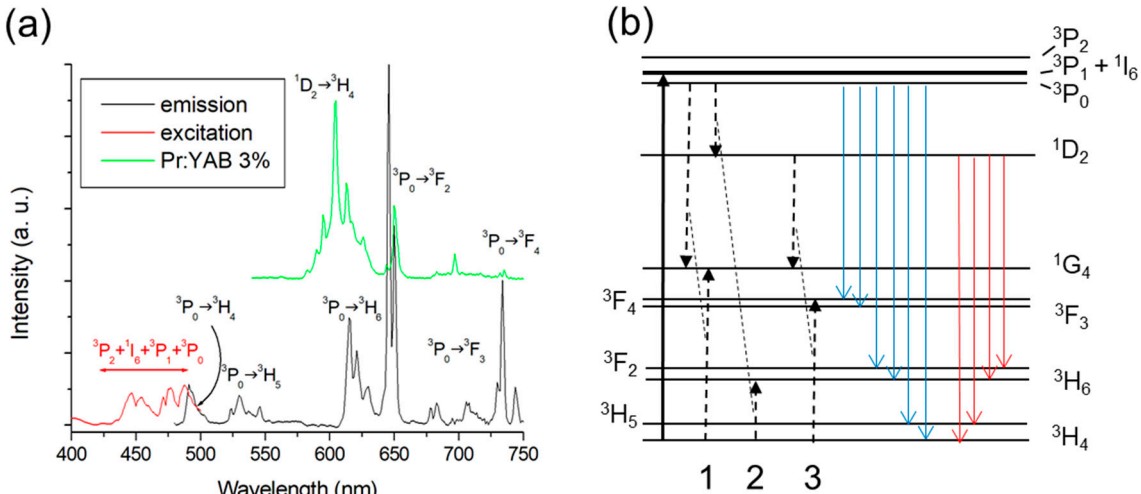

**Figure 2.** (**a**) Excitation (monitored wavelength: 613 nm) and emission (excitation at 450 nm) spectrum of PrAB and of $Pr^{3+}$:YAB. (**b**) Energy levels scheme and de-excitation mechanisms.

The excitation spectra were assigned to the transition from the $^3H_4$ ground state of $Pr^{3+}$ to the excitation levels indicated in Figure 2a. The spectral components were significantly broadened (FWHM~35–40 $cm^{-1}$ for the most intense transitions) and the comparison with the emission of the diluted compound evidenced the complete quenching of the emission from the $^1D_2$ level, usually predominant in the spectra of the diluted materials [25]. These features are both related to the high content of $Pr^{3+}$ ions. The quenching of the $^1D_2$ emission can be ascribed to a cross-relaxation mechanism, $^1D_2$, $^3H_4 \rightarrow ^1G_4$, $^3F_4$ (process one in Figure 2b), which is resonant and then effective. It is even because the $^1D_2$ level is efficiently populated through multi-phonon relaxation from the $^3P_0$ one, with the gap between the two levels being of the order of 3500 $cm^{-1}$ and then bridgeable by 3–4 high energy phonons (the phonon cut-off of YAB is 1070 $cm^{-1}$ [26]), and also through cross-relaxation (process two shown in Figure 2b). In addition, the cross-relaxation mechanism depopulating the $^3P_0$ state ($^3P_0$, $^3H_4 \rightarrow ^1G_4$, $^1G_4$, process three in Figure 2b) was not resonant and less efficient. The combination of these effects meant that only the $^3P_0$ emission was observed in the spectrum of the concentrated crystal.

### 3.2. EuAB

The excitation and emission spectra of EuAB are reported in Figure 3a. They were consistent with the spectra of the diluted Eu:YAB [9] and Eu:GAB [27] and were assigned accordingly. The excitation spectra were assigned to the transitions from the $^7F_0$ ground state of $Eu^{3+}$ to the excited levels indicated in Figure 3a. The spectral features were relatively narrow (FWHM~20 $cm^{-1}$ for the most intense transitions) indicating that the ion-ion interactions were relatively limited. The EuAB spectra were measured by Kellendonk et al. [11] at liquid helium temperature, in order to demonstrate the presence of $Eu^{3+}$ ions in non-regular sites. In their investigation concerning the concentration quenching of the emission, these authors individuated three possible non-radiative mechanisms depleting the $^5D_0$ emitting level: diffusion-limited migration within the regular $Eu^{3+}$ system, energy transfer between ions in regular and non-regular crystallographic sites, migration to quenching centers and transfer to $Mo^{3+}$ ions present as unwanted impurities. The presence of $Eu^{3+}$ in non-regular sites was consistent with the observation of some extra features in the excitation spectra, revealed through comparison with the excitation spectra of the diluted material (see Figure 4). The decay profile of the luminescence, shown in the inset of Figure 3a, is not exponential and can be well reproduced by a two-exponential function with time constants of 112 and 304 μs, whereas the fit based on the diffusive model of Yokota-Tanimoto [28] does not work.

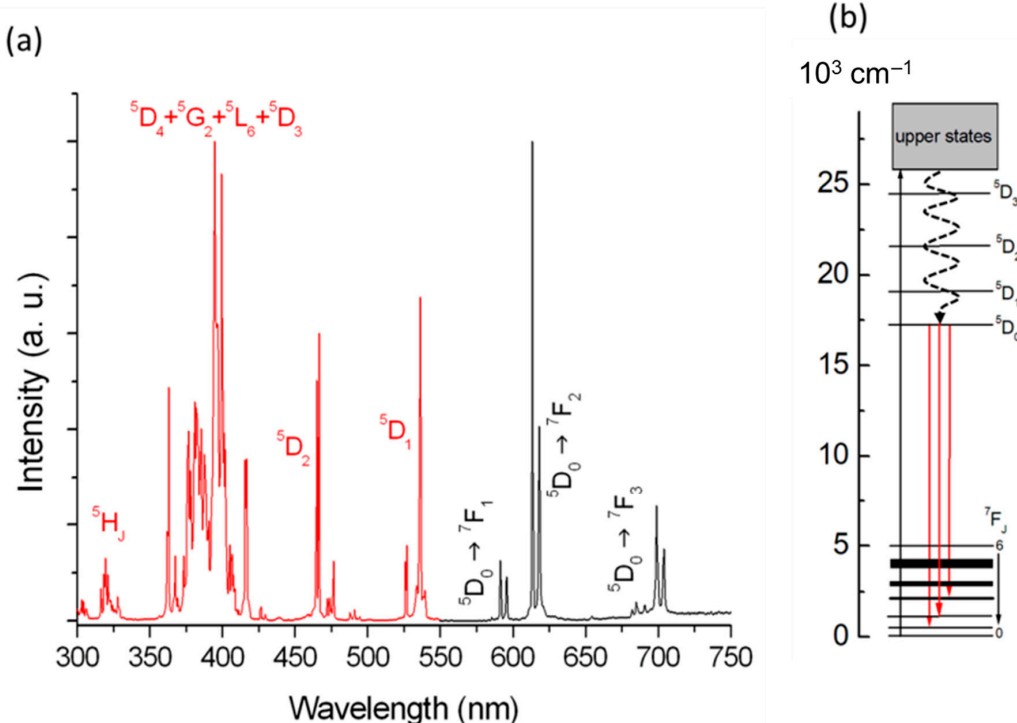

**Figure 3.** (**a**) Excitation (monitored wavelength: 613 nm) and emission (excitation at 394 nm) spectrum of EuAB. (**b**) Energy levels scheme and de-excitation mechanisms.

This behavior is consistent with the presence of different non-radiative processes, resulting in an overall decrease of the quantum yield to about 22% (estimated according to the ratio of the decay times of the concentrated and diluted crystal), with the radiative lifetime of the $^5D_0$ emitting level being 1.35 ms, as shown in Figure 4.

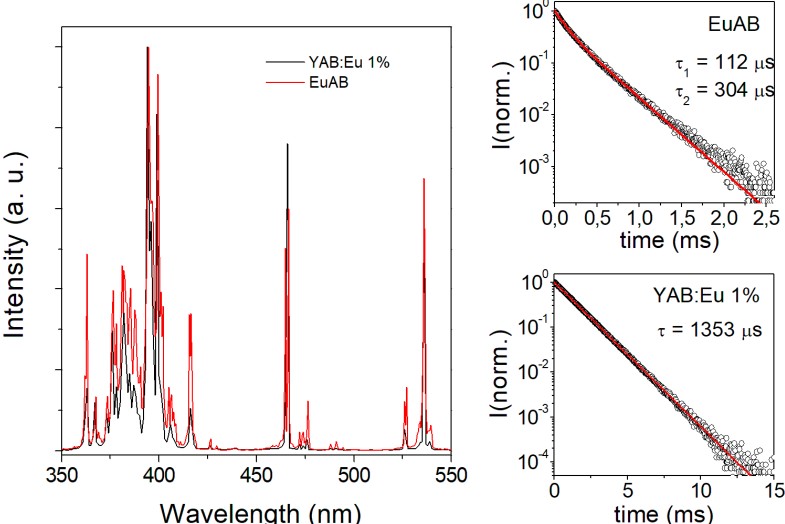

**Figure 4.** Comparison between the excitation spectra of $Eu^{3+}$:YAB and EuAB, and decay profiles of their luminescence.

This value was longer than that reported by Kellendonk (1.12 ms) [11]. Considering that EuAB is a fully concentrated material, its efficiency can be considered relatively high.

### 3.3. TbAB

Apart from the decay profiles, the excitation and emission spectra of TbAB (Figure 5a) were practically identical to those of Tb:YAB 3% and were consistent with previous literature [29,30].

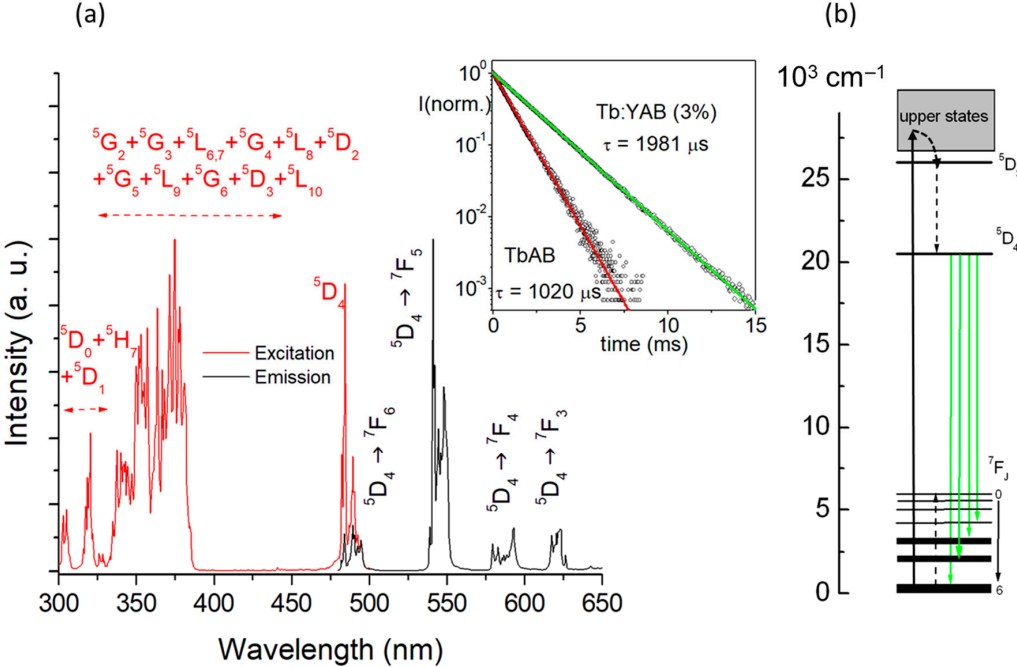

**Figure 5.** (**a**) Excitation (monitored wavelength: 548 nm) and emission (excitation at 375 nm) spectrum of TbAB. Inset: decay profiles of the Tb:YAB and TbAB emission. (**b**) Energy levels scheme and de-excitation mechanisms.

The observed emission transitions originated from the $^5D_4$ excited state, those from the $^5D_3$ one were quenched through the $^5D_3$, $^7F_6 \rightarrow ^5D_4$, $^7F_0$ cross-relaxation mechanism, as shown in Figure 5b. The excitation spectra were attributed to the transition from the $^7F_6$ ground state of $Tb^{3+}$ to the excited levels indicated in Figure 5a. The temporal profiles of the luminescence were a single exponential and the time constant reduced by less than 50% on passing from the concentrated to the diluted material. Together with the absence of any differences in the structure of the spectra, this means that $Tb^{3+}$ occupies a single site in the huntite lattice, in contrast to $Eu^{3+}$. In light of the possible energy transfer mechanisms, the comparison between the EuAB and TbAB spectral properties allowed us to infer that the presence of active ions in non-regular or defective sites plays an important role in reducing the efficiency of the material.

### 3.4. DyAB

To the best of our knowledge, this is the first study to report the excitation and emission spectra, as well as the decay profile, of the luminescence of DyAB (see Figure 6a). The strongest feature was in the yellow region, ascribed to the $^4F_{9/2} \rightarrow ^6H_{13/2}$ transition and interesting for phosphor and laser applications. Similar to the EuAB case, the emission spectrum did not change on passing from the diluted to the fully concentrated compound, whereas the excitation one evidenced a significant broadening and the presence of some extra lines. Thus, it is reasonable to suppose that, in this case, a small part of the doping ions also lie in non-regular sites. It is also important to note that the emitting level can be depleted non-radiatively through a cross-relaxation process ($^4F_{9/2}$, $^6H_{15/2} \rightarrow (^6F_{3/2}$, $^6F_{1/2})$, ($^6H_{9/2}$, $^6F_{11/2})$), shown in Figure 6b. This nearly resonant mechanism accounted for the fast decay of the DyAB emission, whose time constant (2.2 μs) was much shorter than that of the 3% doped Dy:YAB (548 μs).

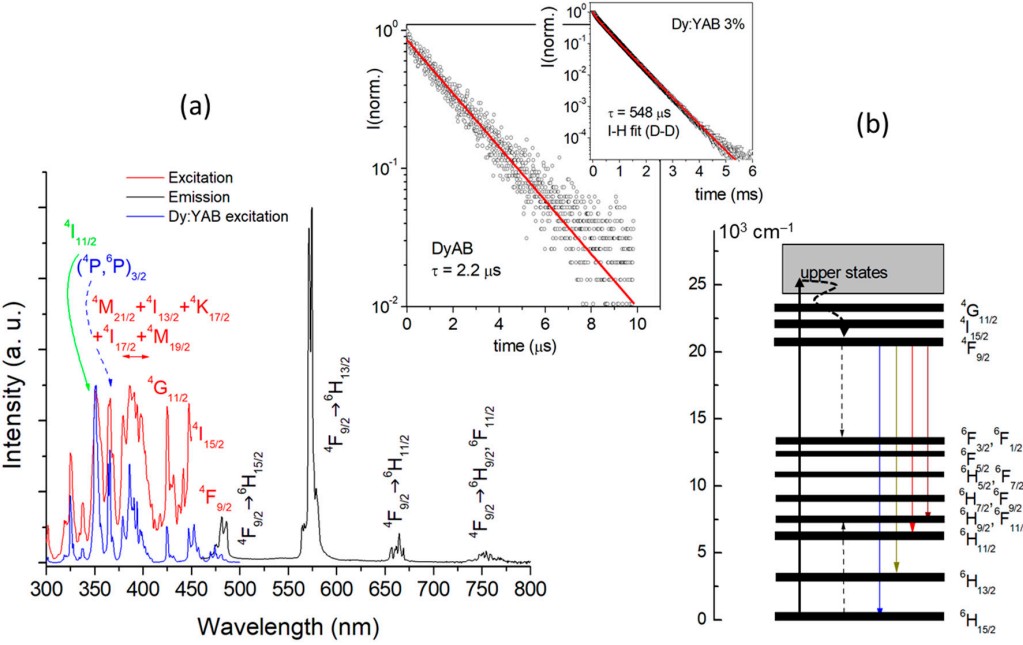

**Figure 6.** (**a**) Excitation (monitored wavelength: 575 nm) and emission (excitation at 351 nm) spectrum of DyAB. Inset: decay profiles of the Dy:YAB and DyAB emission. (**b**) Energy levels scheme and de-excitation mechanisms.

### 3.5. TmAB

The excitation and emission spectra of TmAB are reported in Figure 7a.

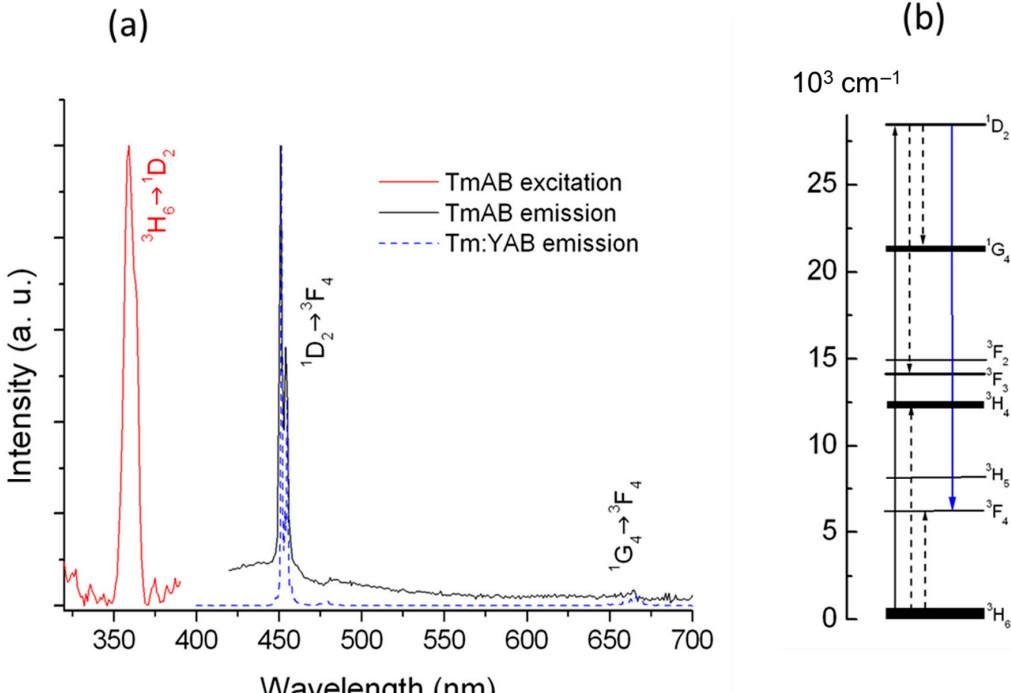

**Figure 7.** (**a**) Excitation (monitored wavelength: 450 nm) and emission (excitation at 358 nm) spectrum of TmAB. (**b**) Energy levels scheme and de-excitation mechanisms.

The luminescence spectrum was measured upon direct excitation into the $^1D_2$ emitting level. It is largely in agreement with the findings reported by Malakhovskii et al. [31], who, however, did not perform excitation or decay time measurements. The corresponding spectrum of Tm:YAB (3%),

is shown for the sake of comparison. The maximum intensity of both spectra is normalized to one. The emission spectrum presented a relatively strong band in the blue region, assigned to the $^1D_2 \rightarrow {}^3F_4$ transition, with other features being only barely appreciable at 480 nm ($^1D_2 \rightarrow {}^3H_5$, nearly absent) and at 665 nm ($^1G_4 \rightarrow {}^3F_4$, weaker in the TmAB spectrum). The blue transition overlapped a broad band whose origin is unknown, but is probably due to impurities ($Ce^{3+}$?). With respect to the emission of the diluted compound, aside from being weaker, it was also relatively broader. The decay profiles were strongly non-exponential in both the diluted and concentrated samples, with the average decay times being 11 µs in the former case and 5 ns in the latter. Considering that the radiative decay time of the $^1D_2$ level is 71 µs [32], it can be concluded that efficient non-radiative processes contributed to depleting the emitting levels. In addition to the energy migration presumably active in TmAB, nearly resonant cross-relaxation processes also occur, as shown in Figure 7b. As a consequence, the concentration quenching, even if incomplete, is rather strong in this material.

### 3.6. YbAB

The optical spectra of $Yb^{3+}$-doped YAB have been extensively investigated [33,34] because of the attractiveness of this material for solid-state laser applications. The spectra, shown in Figure 8a for comparison, were in agreement with previous results. As the intensity of the YbAB emission is rather low, it has been amplified in Figure 8 for the sake of comparison. The spectrum was consistent with that published by Popova et al. [35]. However, the decay profile is reported here for the first time. The observed transitions were broadened mainly because of the strong electron-phonon coupling typical of the $Yb^{3+}$-doped materials. However, the structure of the spectra was different from that of the diluted crystal. In particular, the relative intensity of the excitation and emission bands in the 950–1025 nm range, namely in the vicinity of the 0-0 line at 10188 $cm^{-1}$ (982 nm), was much lower. This is mainly a consequence of reabsorption effects, involving $Yb^{3+}$ ions lying in regular and non-regular sites. In fact, it is known that the optical spectra of $Yb^{3+}$ in YAB are significantly affected by the presence of active ions replacing $Al^{3+}$ or located near to impurities like $Mo^{3+}$ [34–36].

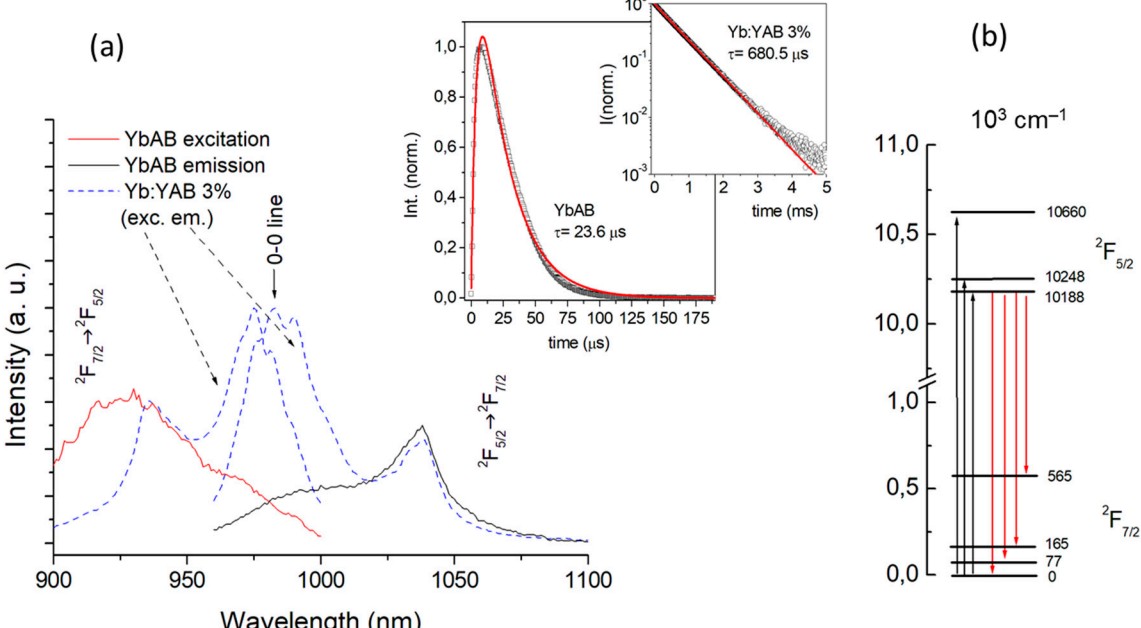

**Figure 8.** (**a**) Excitation (monitored wavelength: 1038 nm) and emission (excitation at 940 nm) spectrum of YbAB. Inset: decay profiles of the Yb:YAB and YbAB emission. (**b**) Energy levels scheme and electronic transitions.

In YbAB, the concentration of these defective species was certainly much higher, as were their effects on the optical features. The build-up in the emission temporal profile and the very short decay constant (23.6 μs versus 680.5 μs for the diluted crystal, see inset of Figure 8) are consistent with the occurrence of energy transfer and migration processes involving active ions located in different environments.

## 4. Discussion and Conclusions

The emission properties of several members of the RAB (R = lanthanide ion) family were investigated. In these materials, the concentration quenching of the luminescence was incomplete, due to the fact that the crystal sites occupied by the active ions were well separated from one another and the energy transfer processes responsible for the quenching were significantly limited. This finding is in agreement with previous studies [11]. The comparison with the spectra of diluted $R^{3+}$:YAB has provided information about the quenching degree and the factors responsible for the creation of non-radiative de-excitation pathways. Efficient cross-relaxation channels reduce the emission efficiency of PrAB and TmAB by more than 90%, and their spectra present features ascribable to defect centers and of difficult attribution. The spectra of EuAB and DyAB are rather similar to those of the diluted crystals, but the efficiency is quite high in the former case (22%) and very low (0.4%) in the latter. The mechanism responsible for the quenching of the EuAB emission is the migration to killer centers, whereas that of the DyAB emission is a cross-relaxation mechanism. The emission of TbAB decreases only by about 50% with respect to the diluted crystal, as a consequence of migration processes. Amongst the studied compounds, this is by far the most efficient. Finally, in the case of YbAB, it must be considered that the absorption and emission processes take place between only two electronic states, which are strongly coupled with the lattice. This implies a significant broadening of the spectral features that favors reabsorption processes. This effect is further enhanced by the presence of $Yb^{3+}$ ions in non-regular lattice sites, whose involvement in migration processes results in the reduction of the emission efficiency to about 3%, with respect that of the diluted material. The above considerations are briefly summarized in Table 3.

**Table 3.** Summary of the luminescence features of the investigated compounds.

| Borate | Emitting Level | Efficiency | Quenching Mechanism |
|---|---|---|---|
| $PrAl_3(BO_3)_4$ | $^3P_0$ | n. d. | Cross-relaxation |
| $EuAl_3(BO_3)_4$ | $^5D_0$ | 22% | Migration |
| $TbAl_3(BO_3)_4$ | $^5D_4$ | 51% | Migration |
| $DyAl_3(BO_3)_4$ | $^4F_{9/2}$ | 0.4% | Cross-relaxation |
| $TmAl_3(BO_3)_4$ | $^1D_2$ | 7% | Cross-relaxation, migration |
| $YbAl_3(BO_3)_4$ | $^2F_{5/2}$ | 4% | Migration, reabsorption |

In the instance that no data were available in the literature or in the absence of agreement between the published data, the quantum efficiencies were evaluated using the decay times of the diluted crystal as a rough estimation of the radiative lifetimes. Consequently, they must be considered as indicative only. As a final consideration, it can be concluded that the concentration quenching in this class of materials is higher when the emitting level is non-radiatively depleted through cross-relaxation mechanisms, which mostly involve regular centers. In contrast, the migration, dependent to a larger extent on the presence of lattice defects, plays a comparatively minor role. The fact that the quenching was incomplete in the fully concentrated RAB crystals is a consequence of the relatively long distance between the active centers in the huntite lattice.

A better characterization of the energy transfer processes involved in the quenching mechanisms could be performed through low temperature spectroscopic measurements, whereas growth experiments in different solvents, like $BaO$-$B_2O_3$ or $Li_2B_4O_7$, could be of help for a more detailed identification of the non-regular active ions. Future work is being planned in these directions.

**Author Contributions:** Conceptualization, spectroscopic measurements and interpretation of the data, manuscript writing: E.C.; Crystal growth and structural characterization, critical reading and review of the manuscript: N.I.L.

**Funding:** This research was supported in part (N.I. Leonyuk) by the RFBR grants ## 18-05-01085_a and 18-29-12091_MK.

**Conflicts of Interest:** The authors declare no conflict of interest.

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
