# Peer review of "Comparative Investigation on the Emission Properties of RAl3(BO3)4 (R = Pr, Eu, Tb, Dy, Tm, Yb) Crystals with the Huntite Structure"

_crystals, doi:10.3390/cryst9010044_

Round 1

Reviewer 1 Report

The authors describe experiments performed on RAl3(BO3)4 compounds containing six different lanthanide elements. The emission properties of the presumably end-member compounds are compared to those of the diluted systems, where the lanthanide is doped into the yttrium analogue. The work is overall of sound quality, although the manuscript does not present the results as clearly as I would like. I have a few specific criticisms:

Chemical characterization is lacking. The unit cell parameters are not sufficient to be sure of chemical purity. This might be expected to have less noticeable effects on the concentrated compound spectra, but it should nevertheless be addressed.

In its current presentation, it is difficult to know which data are new. References for the "diluted" spectra are not given in the figures nor their captions. If both the concentrated and dilute samples were synthesized by the authors then this should be made clear, and characterization would then be much more important.

The figures in general are not presented well. The font size on the "inset" figures is very small, and the choice of layout does not maximize the space afforded by the journal. I would recommend modifying the layout of the figures.

The results section could be better presented using a table of the various data being discussed. For example, columns to facilitate comparison of the efficiencies of the concentrated and diluted samples.

In summary, the work is very dry and poorly motivated. I'm not sure what advance, if any, has been made by carrying out this study. The spectra appear to be assigned correctly, but none of this is new. I would recommend working on the results section to provide clearer context for the results of this study.

Author Response

See uploaded file.

Reviewer 2 Report

All crystals were grown by the flux technique. Please, indicate typical dimensions of grown crystals and samples you studied.

One of the problems of crystals grown by the flux technique is diffusion of flux component into crystal lattice. Did you perform chemical analysis of your crystals? What are concentrations of K and Mo in the crystals? Is there dependence of K and Mo content on RE cation?

Luminescent and kinetic characteristics of crystals containing RE3+ ions are sensitive to a local surrounding (defects and uncontrolled impurities) in a crystal lattice. Can you tell anything about typical defects in huntite family crystals grown by the flux technique?

Author Response

See uploaded file.
